# Conditional Random Field Autoencoders
# for Unsupervised Structured Prediction

**Waleed Ammar**  **Chris Dyer**  **Noah A. Smith**
School of Computer Science
Carnegie Mellon University
Pittsburgh, PA 15213, USA
{wammar,cdyer,nasmith}@cs.cmu.edu

## Abstract

We introduce a framework for unsupervised learning of structured predictors with overlapping, global features. Each input's latent representation is predicted conditional on the observed data using a feature-rich conditional random field (CRF). Then a reconstruction of the input is (re)generated, conditional on the latent structure, using a generative model which factorizes similarly to the CRF. The autoencoder formulation enables efficient exact inference without resorting to unrealistic independence assumptions or restricting the kinds of features that can be used. We illustrate connections to traditional autoencoders, posterior regularization, and multi-view learning. We then show competitive results with instantiations of the framework for two canonical tasks in natural language processing: part-of-speech induction and bitext word alignment, and show that training the proposed model can be substantially more efficient than a comparable feature-rich baseline.

## 1 Introduction

Conditional random fields [24] are used to model structure in numerous problem domains, including natural language processing (NLP), computational biology, and computer vision. They enable efficient inference while incorporating rich features that capture useful domain-specific insights. Despite their ubiquity in supervised settings, CRFs—and, crucially, the insights about effective feature sets obtained by developing them—play less of a role in *unsupervised* structure learning, a problem which traditionally requires jointly modeling observations and the latent structures of interest. For unsupervised structured prediction problems, less powerful models with stronger independence assumptions are standard.[1] This state of affairs is suboptimal in at least three ways: (i) adhering to inconvenient independence assumptions when designing features is limiting—we contend that effective feature engineering is a crucial mechanism for incorporating inductive bias in unsupervised learning problems; (ii) features and their weights have different semantics in joint and conditional models (see §3.1); and (iii) modeling the generation of high-dimensional observable data with feature-rich models is computationally challenging, requiring expensive marginal inference in the inner loop of iterative parameter estimation algorithms (see §3.1).

Our approach leverages the power and flexibility of CRFs in unsupervised learning without sacrificing their attractive computational properties or changing the semantics of well-understood feature sets. Our approach replaces the standard joint model of observed data and latent structure with a two-layer **conditional random field autoencoder** that first generates latent structure with a CRF (conditional on the observed data) and then (re)generates the observations conditional on just the predicted structure. For the reconstruction model, we use distributions which offer closed-form maximum

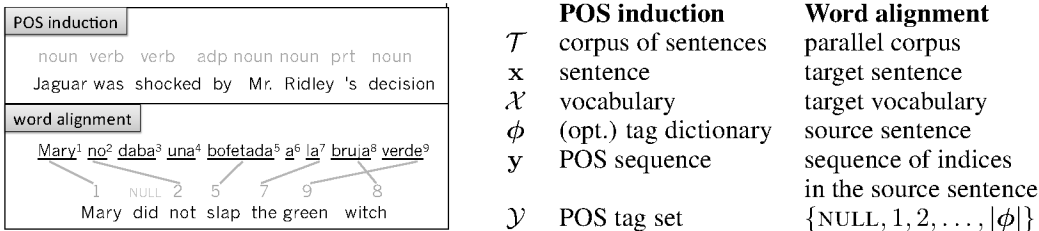

| | **POS induction** | **Word alignment** |
|---|---|---|
| $\mathcal{T}$ | corpus of sentences | parallel corpus |
| $\mathbf{x}$ | sentence | target sentence |
| $\mathcal{X}$ | vocabulary | target vocabulary |
| $\phi$ | (opt.) tag dictionary | source sentence |
| $\mathbf{y}$ | POS sequence | sequence of indices in the source sentence |
| $\mathcal{Y}$ | POS tag set | $\{\text{NULL}, 1, 2, \ldots, |\phi|\}$ |

Figure 1: Left: Examples of structured observations (in black), hidden structures (in gray), and side information (underlined). Right: Model variables for POS induction (§A.1) and word alignment (§A.2). A parallel corpus consists of pairs of sentences ("source" and "target").

likelihood estimates (§2). The proposed architecture provides several mechanisms for encouraging the learner to use its latent variables to find intended (rather than common but irrelevant) correlations in the data. First, hand-crafted feature representations—engineered using knowledge about the problem—provide a key mechanism for incorporating inductive bias. Second, by reconstructing a transformation of the structured input. Third, it is easy to simultaneously learn from labeled and unlabeled examples in this architecture, as we did in [26]. In addition to the modeling flexibility, our approach is computationally efficient in a way achieved by no other unsupervised, feature-based model to date: under a set of mild independence assumptions regarding the reconstruction model, per-example inference required for learning in a CRF autoencoder is no more expensive than in a supervised CRF with the same independence assumptions.

In the next section we describe the modeling framework, then review related work and show that, although our model admits more powerful features, the inference required for learning is simpler (§3). We conclude with experiments showing that the proposed model achieves state-of-the-art performance on two unsupervised tasks in NLP: POS induction and word alignment, and find that it is substantially more efficient than MRFs using the same feature set (§4).

## 2 Conditional Random Field Autoencoder

Given a training set $\mathcal{T}$ of observations (e.g., sentences or pairs of sentences that are translationally equivalent), consider the problem of inducing the hidden structure in each observation. Examples of hidden structures include shallow syntactic properties (part-of-speech or POS tags), correspondences between words in translation (word alignment), syntactic parses and morphological analyses.

**Notation.** Let each observation be denoted $\mathbf{x} = \langle x_1, \ldots, x_{|\mathbf{x}|} \rangle \in \mathcal{X}^{|\mathbf{x}|}$, a variable-length tuple of discrete variables, $x \in \mathcal{X}$. The hidden variables $\mathbf{y} = \langle y_1, \ldots, y_{|\mathbf{y}|} \rangle \in \mathcal{Y}^{|\mathbf{y}|}$ form a tuple whose length is determined by $\mathbf{x}$, also taking discrete values.[2] We assume that $|\mathcal{Y}|^{|\mathbf{y}|} \ll |\mathcal{X}|^{|\mathbf{x}|}$, which is typical in structured prediction problems. Fig. 1 (right) instantiates $\mathbf{x}$, $\mathcal{X}$, $\mathbf{y}$, and $\mathcal{Y}$ for two NLP tasks.

Our model introduces a new observed variable, $\hat{\mathbf{x}} = \langle \hat{x}_1, \ldots, \hat{x}_{|\hat{\mathbf{x}}|} \rangle$. In the basic model in Fig. 2 (left), $\hat{\mathbf{x}}$ is a copy of $\mathbf{x}$. The intuition behind this model is that a good hidden structure should be a likely *encoding* of observed data and should permit *reconstruction* of the data with high probability.

**Sequential latent structure.** In this paper, we focus on *sequential* latent structures with first-order Markov properties, i.e., $y_i \perp y_j \mid \{y_{i-1}, y_{i+1}\}$, as illustrated in Fig. 2 (right). This class of latent structures is a popular choice for modeling a variety of problems such as human action recognition [47], bitext word alignment [6, 45, 4], POS tagging [28, 20], acoustic modeling [19], gene finding [27], and transliteration [36], among others. Importantly, we make no assumptions about conditional independence between any $y_i$ and $\mathbf{x}$.

Eq. 1 gives the parameteric form of the model for sequence labeling problems. $\lambda$ and $\theta$ are the parameters of the encoding and reconstruction models, respectively. $\mathbf{g}$ is a vector of clique-local

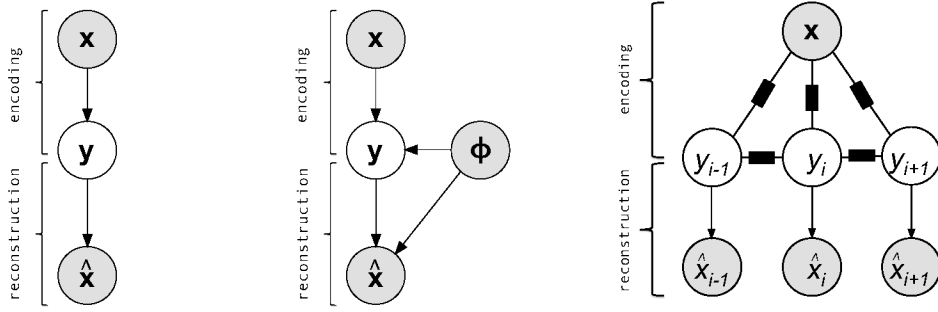

Figure 2: Graphical model representations of CRF autoencoders. Left: the basic autoencoder model where the observation $\mathbf{x}$ generates the hidden structure $\mathbf{y}$ (encoding), which then generates $\hat{\mathbf{x}}$ (reconstruction). Center: side information ($\phi$) is added. Right: a factor graph showing first-order Markov dependencies among elements of the hidden structure $\mathbf{y}$.

feature functions.[3]

$$p_{\boldsymbol{\lambda},\boldsymbol{\theta}}(\hat{\mathbf{x}} \mid \mathbf{x}) = \sum_{\mathbf{y}} p_{\boldsymbol{\lambda}}(\mathbf{y} \mid \mathbf{x}) p_{\boldsymbol{\theta}}(\hat{\mathbf{x}} \mid \mathbf{y}) = \sum_{\mathbf{y} \in \mathcal{Y}^{|\mathbf{x}|}} \frac{e^{\sum_{i=1}^{|\mathbf{x}|} \boldsymbol{\lambda}^{\top} \mathbf{g}(\mathbf{x}, y_i, y_{i-1}, i)}}{\sum_{\mathbf{y}' \in \mathcal{Y}^{|\mathbf{x}|}} e^{\sum_{i=1}^{|\mathbf{x}|} \boldsymbol{\lambda}^{\top} \mathbf{g}(\mathbf{x}, y_i', y_{i-1}', i)}} \prod_{i=1}^{|\mathbf{x}|} p_{\boldsymbol{\theta}}(\hat{x}_i \mid y_i)$$

$$= \frac{\sum_{\mathbf{y} \in \mathcal{Y}^{|\mathbf{x}|}} \exp\left(\sum_{i=1}^{|\mathbf{x}|} \log \theta_{\hat{x}_i \mid y_i} + \boldsymbol{\lambda}^{\top} \mathbf{g}(\mathbf{x}, y_i, y_{i-1}, i)\right)}{\sum_{\mathbf{y}' \in \mathcal{Y}^{|\mathbf{x}|}} \exp \sum_{i=1}^{|\mathbf{x}|} \boldsymbol{\lambda}^{\top} \mathbf{g}(\mathbf{x}, y_i', y_{i-1}', i)} \tag{1}$$

**Encoding and reconstruction.** We model the *encoding* part with a CRF, which allows us to exploit features with global scope in the structured observation $\mathbf{x}$, while keeping exact inference tractable (since the model does not generate $\mathbf{x}$, only conditions on it). The *reconstruction* part, on the other hand, grounds the model by generating a copy of the structured observations. We use a simple categorical (i.e., multinomial) distribution to independently generate $\hat{x}_i$ given $y_i$.[4] Fig. 2 (right) is an instance of the model for POS induction with a sequential latent structure; each $\hat{x}_i$ is generated from $p_{\boldsymbol{\theta}}(\hat{x}_i \mid y_i)$.

We emphasize the importance of allowing modelers to define intuitive feature templates in a flexible manner. The need to efficiently add inductive bias via feature engineering has been the primary drive for developing CRF autoencoders. For example, morphology, word spelling information, and other linguistic knowledge encoded as features were shown to improve POS induction [38], word alignment [14], and other unsupervised learning problems. The proposed model enables the modeler to define such features at a lower computational cost *and* enables more expressive features with global scope in the structured input. For example, we found that using predictions of other models as features is an effective method for model combination in unsupervised word alignment tasks, and found that conjoining sub-word-level features of consecutive words helps disambiguate their POS labels (see Appendix A for experimental details).

**Extension: side information.** Our model can be easily extended to condition on more context in the encoding part, the reconstruction part, or in both parts. Let $\phi$ represent *side information*: additional context which we may condition on in the encoding part, the reconstruction part, or both. In our running example, side information could represent the set of possible tags for each word, a common form of weak "supervision" shown to help unsupervised POS learners. In word alignment, where $y_i = j$ indicates that $x_i$ translates to the $j$th source token, we treat the source sentence as side information, making its word forms available for feature extraction.

**Extension: partial reconstruction.** In our running POS example, the reconstruction model $p_{\boldsymbol{\theta}}(\hat{x}_i \mid y_i)$ defines a distribution over words given tags. Because word distributions are heavy-tailed, estimating such a distribution reliably is quite challenging. Our solution is to define a function $\pi : \mathcal{X} \to \hat{\mathcal{X}}$ such that $|\hat{\mathcal{X}}| \ll |\mathcal{X}|$, and let $\hat{x}_i = \pi(x_i)$ be a deterministic transformation of the original structured observation. We can add indirect supervision by defining $\pi$ such that it represents observed information relevant to the latent structure of interest. For example, we found reconstructing Brown clusters [5] of tokens instead of their surface forms to improve POS induction. Other possible reconstructions include word embeddings, morphological and spelling features of words.

**More general graphs.** We presented the CRF autoencoder in terms of sequential Markovian assumptions for ease of exposition; however, this framework can be used to model arbitrary hidden structures. For example, instantiations of this model can be used for unsupervised learning of parse trees [21], semantic role labels [42], and coreference resolution [35] (in NLP), motif structures [1] in computational biology, and object recognition [46] in computer vision. The requirements for applying the CRF autoencoder model are:

- An encoding discriminative model defining $p_{\boldsymbol{\lambda}}(\mathbf{y} \mid \mathbf{x}, \boldsymbol{\phi})$. The encoder may be any model family where *supervised* learning from $\langle \mathbf{x}, \mathbf{y} \rangle$ pairs is efficient.
- A reconstruction model that defines $p_{\boldsymbol{\theta}}(\hat{\mathbf{x}} \mid \mathbf{y}, \boldsymbol{\phi})$ such that inference over $\mathbf{y}$ given $\langle \mathbf{x}, \hat{\mathbf{x}} \rangle$ is efficient.
- The independencies among $\mathbf{y} \mid \mathbf{x}, \hat{\mathbf{x}}$ are not strictly weaker than those among $\mathbf{y} \mid \mathbf{x}$.

## 2.1 Learning & Inference

Model parameters are selected to maximize the regularized conditional log likelihood of reconstructed observations $\hat{\mathbf{x}}$ given the structured observation $\mathbf{x}$:

$$\ell\ell(\boldsymbol{\lambda}, \boldsymbol{\theta}) = R_1(\boldsymbol{\lambda}) + R_2(\boldsymbol{\theta}) + \sum_{(\mathbf{x}, \hat{\mathbf{x}}) \in \mathcal{T}} \log \sum_{\mathbf{y}} p_{\boldsymbol{\lambda}}(\mathbf{y} \mid \mathbf{x}) \times p_{\boldsymbol{\theta}}(\hat{\mathbf{x}} \mid \mathbf{y}) \tag{2}$$

We apply block coordinate descent, alternating between maximizing with respect to the CRF parameters ($\boldsymbol{\lambda}$-step) and the reconstruction parameters ($\boldsymbol{\theta}$-step). Each $\boldsymbol{\lambda}$-step applies one or two iterations of a gradient-based convex optimizer.[5] The $\boldsymbol{\theta}$-step applies one or two iterations of EM [10], with a closed-form solution in the M-step in each EM iteration. The independence assumptions among $\mathbf{y}$ make the marginal inference required in both steps straightforward; we omit details for space.

In the experiments below, we apply a squared $L_2$ regularizer for the CRF parameters $\boldsymbol{\lambda}$, and a symmetric Dirichlet prior for categorical parameters $\boldsymbol{\theta}$.

The asymptotic runtime complexity of each block coordinate descent iteration, assuming the first-order Markov dependencies in Fig. 2 (right), is:

$$O\left(|\boldsymbol{\theta}| + |\boldsymbol{\lambda}| + |\mathcal{T}| \times |\mathbf{x}|_{max} \times |\mathcal{Y}|_{max} \times (|\mathcal{Y}|_{max} \times |F_{y_{i-1}, y_i}| + |F_{\mathbf{x}, y_i}|)\right) \tag{3}$$

where $F_{y_{i-1}, y_i}$ are the active "label bigram" features used in $\langle y_{i-1}, y_i \rangle$ factors, $F_{\mathbf{x}, y_i}$ are the active emission-like features used in $\langle \mathbf{x}, y_i \rangle$ factors. $|\mathbf{x}|_{max}$ is the maximum length of an observation sequence. $|\mathcal{Y}|_{max}$ is the maximum cardinality[6] of the set of possible assignments of $y_i$.

After learning the $\boldsymbol{\lambda}$ and $\boldsymbol{\theta}$ parameters of the CRF autoencoder, test-time predictions are made using maximum a posteriori estimation, conditioning on both observations and reconstructions, i.e., $\hat{\mathbf{y}}_{\text{MAP}} = \arg\max_{\mathbf{y}} p_{\boldsymbol{\lambda}, \boldsymbol{\theta}}(\mathbf{y} \mid \mathbf{x}, \hat{\mathbf{x}})$.

## 3 Connections To Previous Work

This work relates to several strands of work in unsupervised learning. Two broad types of models have been explored that support unsupervised learning with flexible feature representations. Both are

fully generative models that define joint distributions over $\mathbf{x}$ and $\mathbf{y}$. We discuss these "undirected" and "directed" alternatives next, then turn to less closely related methods.

### 3.1 Existing Alternatives for Unsupervised Learning with Features

**Undirected models.** A Markov random field (MRF) encodes the joint distribution through local potential functions parameterized using features. Such models "normalize globally," requiring during training the calculation of a partition function summing over all possible inputs and outputs. In our notation:

$$Z(\boldsymbol{\theta}) = \sum_{\mathbf{x} \in \mathcal{X}^*} \sum_{\mathbf{y} \in \mathcal{Y}^{|\mathbf{x}|}} \exp \boldsymbol{\lambda}^\top \bar{\mathbf{g}}(\mathbf{x}, \mathbf{y}) \tag{4}$$

where $\bar{\mathbf{g}}$ collects all the local factorization by cliques of the graph, for clarity. The key difficulty is in the summation over all possible observations. Approximations have been proposed, including contrastive estimation, which sums over subsets of $\mathcal{X}^*$ [38, 43] (applied variously to POS learning by Haghighi and Klein [18] and word alignment by Dyer et al. [14]) and noise contrastive estimation [30].

**Directed models.** The directed alternative avoids the global partition function by factorizing the joint distribution in terms of locally normalized conditional probabilities, which are parameterized in terms of features. For unsupervised sequence labeling, the model was called a "feature HMM" by Berg-Kirkpatrick et al. [3]. The local emission probabilities $p(x_i \mid y_i)$ in a first-order HMM for POS tagging are reparameterized as follows (again, using notation close to ours):

$$p_{\boldsymbol{\lambda}}(x_i \mid y_i) = \frac{\exp \boldsymbol{\lambda}^\top \mathbf{g}(x_i, y_i)}{\sum_{x \in \mathcal{X}} \exp \boldsymbol{\lambda}^\top \mathbf{g}(x, y_i)} \tag{5}$$

The features relating hidden to observed variables must be local within the factors implied by the directed graph. We show below that this locality restriction excludes features that are useful (§A.1).

Put in these terms, the proposed autoencoding model is a hybrid directed-undirected model.

**Asymptotic Runtime Complexity of Inference.** The models just described cannot condition on arbitrary amounts of $\mathbf{x}$ without increasing inference costs. Despite the strong independence assumptions of those models, the computational complexity of inference required for learning with CRF autoencoders is better (§2.1).

Consider learning the parameters of an undirected model by maximizing likelihood of the observed data. Computing the gradient for a training instance $\mathbf{x}$ requires time

$$\mathrm{O}\left(|\boldsymbol{\lambda}| + |\mathcal{T}| \times |\mathbf{x}| \times |\mathcal{Y}| \times (|\mathcal{Y}| \times |F_{y_{i-1}, y_i}| + |\mathcal{X}| \times |F_{x_i, y_i}|)\right),$$

where $F_{x_i - y_i}$ are the emission-like features used in an arbitrary assignment of $x_i$ and $y_i$. When the multiplicative factor $|\mathcal{X}|$ is large, inference is slow compared to CRF autoencoders.

Inference in *directed models* is faster than in undirected models, but still slower than CRF autoencoder models. In directed models [3], each iteration requires time

$$\mathrm{O}\left(|\boldsymbol{\lambda}| + |\mathcal{T}| \times |\mathbf{x}| \times |\mathcal{Y}| \times (|\mathcal{Y}| \times |F_{y_{i-1}, y_i}| + |F_{x_i, y_i}|) + |\boldsymbol{\theta}'| \times \max(|F_{y_{i-1}, y_i}|, |F_{\mathcal{X}, y_i}|)\right),$$

where $F_{x_i, y_i}$ are the active emission features used in an arbitrary assignment of $x_i$ and $y_i$, $F_{\mathcal{X}, y_i}$ is the union of all emission features used with an arbitrary assignment of $y_i$, and $\boldsymbol{\theta}'$ are the local emission and transition probabilities. When $|\mathcal{X}|$ is large, the last term $|\boldsymbol{\theta}'| \times \max(|F_{y_{i-1}, y_i}|, |F_{\mathcal{X}, y_i}|)$ can be prohibitively large.

### 3.2 Other Related Work

The proposed CRF autoencoder is more distantly related to several important ideas in less-than-supervised learning.

**Autoencoders and other "predict self" methods.** Our framework borrows its general structure, Fig. 2 (left), as well as its name, from *neural network* autoencoders. The goal of neural autoencoders has been to learn feature representations that improve generalization in otherwise supervised learning problems [44, 8, 39]. In contrast, the goal of CRF autoencoders is to learn specific *interpretable* regularities of interest.[7] It is not clear how neural autoencoders could be used to learn the latent structures that CRF autoencoders learn, without providing supervised training examples. Stoyanov et al. [40] presented a related approach for discriminative graphical model learning, including features and latent variables, based on backpropagation, which could be used to instantiate the CRF autoencoder.

Daumé III [9] introduced a reduction of an unsupervised problem instance to a series of single-variable supervised classifications. The first series of these construct a latent structure $\mathbf{y}$ given the entire $\mathbf{x}$, then the second series reconstruct the input. The approach can make use of any supervised learner; if feature-based probabilistic models were used, a $|\mathcal{X}|$ summation (akin to Eq. 5) would be required. On unsupervised POS induction, this approach performed on par with the undirected model of Smith and Eisner [38].

Minka [29] proposed cascading a generative model and a discriminative model, where class labels (to be predicted at test time) are marginalized out in the generative part first, and then (re)generated in the discriminative part. In CRF autoencoders, observations (available at test time) are conditioned on in the discriminative part first, and then (re)generated in the generative part.

**Posterior regularization.** Introduced by Ganchev et al. [16], posterior regularization is an effective method for specifying constraint on the posterior distributions of the latent variables of interest; a similar idea was proposed independently by Bellare et al. [2]. For example, in POS induction, every sentence might be expected to contain at least one verb. This is imposed as a soft constraint, i.e., a feature whose expected value under the model's posterior is constrained. Such expectation constraints are specified directly by the domain-aware model designer.[8] The approach was applied to unsupervised POS induction, word alignment, and parsing. Although posterior regularization was applied to directed feature-less generative models, the idea is orthogonal to the model family and can be used to add more inductive bias for training CRF autoencoder models.

## 4  Evaluation

We evaluate the effectiveness of CRF autoencoders for learning from unlabeled examples in POS induction and word alignment. We defer the detailed experimental setup to Appendix A.

**Part-of-Speech Induction Results.** Fig. 3 compares predictions of the CRF autoencoder model in seven languages to those of a featurized first-order HMM model [3] and a standard (feature-less) first-order HMM, using V-measure [37] (higher is better). First, note the large gap between both feature-rich models on the one hand, and the feature-less HMM model on the other hand. Second, note that CRF autoencoders outperform featurized HMMs in all languages, except Italian, with an average relative improvement of 12%.

These results provide empirical evidence that feature engineering is an important source of inductive bias for unsupervised structured prediction problems. In particular, we found that using Brown cluster reconstructions and specifying features which span multiple words significantly improve the performance. Refer to Appendix A for more analysis.

**Bitext Word Alignment Results.** First, we consider an intrinsic evaluation on a Czech-English dataset of manual alignments, measuring the alignment error rate (AER; [32]). We also perform an

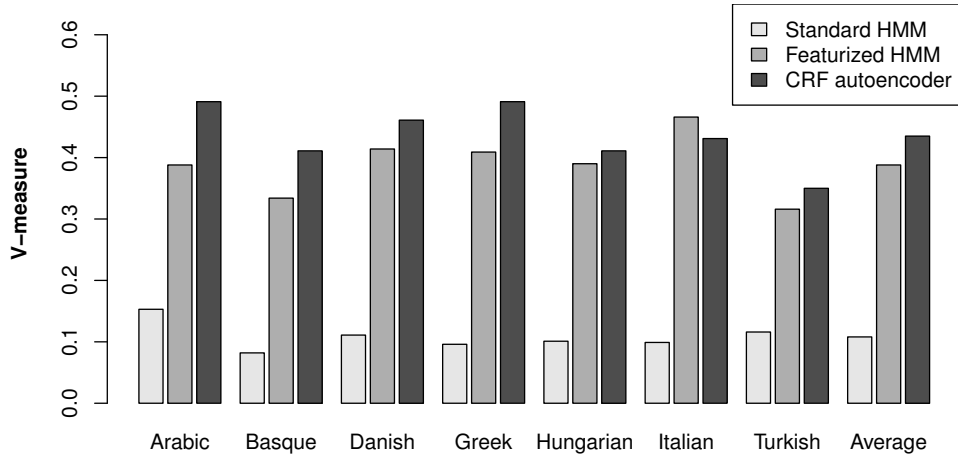

Figure 3: V-measure [37] of induced parts of speech in seven languages. The CRF autoencoder with features spanning multiple words and with Brown cluster reconstructions achieves the best results in all languages but Italian, closely followed by the feature-rich HMM of Berg-Kirkpatrick et al. [3]. The standard multinomial HMM consistently ranks last.

| direction | fast_align | model 4 | auto | | pair | fast_align | model 4 | auto |
|---|---|---|---|---|---|---|---|---|
| forward | 27.7 | 31.5 | **27.5** | | cs-en | $15.2_{\pm0.3}$ | $15.3_{\pm0.1}$ | $\mathbf{15.5}_{\pm0.1}$ |
| reverse | 25.9 | 24.1 | **21.1** | | ur-en | $20.0_{\pm0.6}$ | $20.1_{\pm0.6}$ | $\mathbf{20.8}_{\pm0.5}$ |
| symmetric | 25.2 | 22.2 | **19.5** | | zh-en | $\mathbf{56.9}_{\pm1.6}$ | $56.7_{\pm1.6}$ | $56.1_{\pm1.7}$ |

Table 1: Left: AER results (%) for Czech-English word alignment. Lower values are better. . Right: Bleu translation quality scores (%) for Czech-English, Urdu-English and Chinese-English. Higher values are better. .

extrinsic evaluation of translation quality in three language pairs, using case-insensitive Bleu [33] of a machine translation system (cdec[9] [13]) built using the word alignment predictions of each model.

AER for variants of each model (forward, reverse, and symmetrized) are shown in Table 1 (left). Our model significantly outperforms both baselines. Bleu scores on the three language pairs are shown in Table 1; alignments obtained with our CRF autoencoder model improve translation quality of the Czech-English and Urdu-English translation systems, but not of Chinese-English. This is unsurprising, given that Chinese orthography does not use letters, so that source-language spelling and morphology features our model incorporates introduce only noise here. Better feature engineering, or more data, is called for.

We have argued that the feature-rich CRF autoencoder will scale better than its feature-rich alternatives. Fig. 5 (in Appendix A.2) shows the average per-sentence inference runtime for the CRF autoencoder compared to exact inference in an MRF [14] with a similar feature set, as a function of the number of sentences in the corpus. For CRF autoencoders, the average inference runtime grows slightly due to the increased number of parameters, while it grows substantially with vocabulary size in MRF models [14].[10]

## 5  Conclusion

We have presented a general and scalable framework to learn from unlabeled examples for structured prediction. The technique allows features with global scope in observed variables with favorable asymptotic inference runtime. We achieve this by embedding a CRF as the encoding model in the

input layer of an autoencoder, and reconstructing a transformation of the input at the output layer using simple categorical distributions. The key advantages of the proposed model are scalability and modeling flexibility. We applied the model to POS induction and bitext word alignment, obtaining results that are competitive with the state of the art on both tasks.

### Acknowledgments

We thank Brendan O'Connor, Dani Yogatama, Jeffrey Flanigan, Manaal Faruqui, Nathan Schneider, Phil Blunsom and the anonymous reviewers for helpful suggestions. We also thank Taylor Berg-Kirkpatrick for providing his implementation of the POS induction baseline, and Phil Blunsom for sharing POS induction evaluation scripts. This work was sponsored by the U.S. Army Research Laboratory and the U.S. Army Research Office under contract/grant number W911NF-10-1-0533. The statements made herein are solely the responsibility of the authors.

## Footnotes

[1] For example, a first-order hidden Markov model requires that $y_i \perp x_{i+1} \mid y_{i+1}$ for a latent sequence $\mathbf{y} = \langle y_1, y_2, \ldots \rangle$ generating $\mathbf{x} = \langle x_1, x_2, \ldots \rangle$, while a first-order CRF allows $y_i$ to directly depend on $x_{i+1}$.

[2] In the interest of notational simplicity, we conflate random variables with their values.

[3]We define $y_0$ to be a fixed "start" tag.

[4]Other possible parameterizations of the reconstruction model we would like to experiment with include a multivariate Gaussian for generating word embeddings and a naïve Bayes model for generating individual features of a word independently conditioned on the corresponding label.

[5]We experimented with AdaGrad [12] and L-BFGS. When using AdaGrad, we accummulate the gradient vectors across block coordinate ascent iterations.

[6]In POS induction, $|\mathcal{Y}|$ is a constant, the number of syntactic classes which we configure to 12 in our experiments. In word alignment, $|\mathcal{Y}|$ is the size of the source sentence plus one, therefore $|\mathcal{Y}|_{max}$ is the maximum length of a source sentence in the bitext corpus.

[7]This is possible in CRF autoencoders due to the interdependencies among variables in the hidden structure and the manually specified feature templates which capture the relationship between observations and their hidden structures.

[8]In a semi-supervised setting, when some labeled examples of the hidden structure are available, Druck and McCallum [11] used labeled examples to estimate desirable expected values. We leave semi-supervised applications of CRF autoencoders to future work; see also Suzuki and Isozaki [41].

[9]http://www.cdec-decoder.org/

[10]We only compare runtime, instead of alignment quality, because retraining the MRF model with exact inference was too expensive.

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
