[Supplementary Material]

# A Experimental Details

In this section, we describe our experimental setup for part of speech induction and bitext word alignment in some detail.

## A.1 Part-of-Speech Induction Experimental Setup

The first task, part-of-speech (POS) induction, is a classic NLP problem which aims at discovering syntactic classes of tokens in a monolingual corpus, with a predefined number of classes. An example of a POS-tagged English sentence is in Fig. 1.

**Data.** We use the plain text from CoNLL-X [7] and CoNLL 2007 [31] training data in seven languages to train the models: Arabic, Basque, Danish, Greek, Hungarian, Italian and Turkish. For evaluation, we obtain gold-standard POS tags by deterministically mapping the language-specific POS tags from the shared task training data to the corresponding universal POS tag set [34].[11]

**Setup.** We configure our model (as well as baseline models) to induce $|\mathcal{Y}| = 12$ classes. We use zero initialization of the CRF parameters, and initialize the reconstruction model parameters with a basic first-order HMM. In each block-coordinate ascent iteration, we run one L-BFGS iteration (including a line search) to optimize $\lambda$, followed by one EM iteration to optimize $\theta$. Training converges after 70 block-coordinate ascent iterations.

**Evaluation.** Since we do not use a tagging dictionary, the word classes induced by our model are not identifiable. We use two cluster evaluation metrics commonly used for POS induction: (i) many-to-one [20], which infers a mapping across the syntactic clusters in the gold vs. predicted labels (higher is better); and (ii) V-measure [37], an entropy-based metric which explicitly measures the homogeneity and completeness of predicted clusters (again, higher is better).

**CRF Autoencoder Model Instantiation.** Table 1 (right) describes the symbols and variables we use in context of the POS induction problem. We use a first-order linear CRF for the encoding part with the following feature templates:

- $\langle y_i, y_{i-1} \rangle, \forall i$
- $\langle y_i, \text{sub}_j(x_i) \rangle, \forall i, j$
- $\langle y_i, \text{sub}_j(x_i), \text{sub}_k(x_{i-1}) \rangle, \forall i, j, k$
- $\langle y_i, \text{sub}_j(x_i), \text{sub}_k(x_{i+1}) \rangle, \forall i, j, k$

Where $\text{sub}_j(x_i)$ is one of the following sub-word-level feature percepts:

- Prefixes and suffixes of lengths two and three, iff the affix appears in more than 0.02% of all word types,
- Whether the word contains a digit,
- Whether the word contains a hyphen,
- Whether the word starts with a capital letter,
- Word shape features which map sequences of the same character classes into a single character (e.g., 'McDonalds' → 'AaAa', '-0.5' → '#0#0'),
- The lowercased word, iff it appears more than 100 times in the corpus.

In the reconstruction model, we generate the Brown cluster of a word [5],[12] conditioned on the POS tag using a categorical (i.e., multinomial) distribution. No side information is used in this model instantiation. Predictions are the best value of the latent structure according to the posterior $p(\mathbf{y} \mid \mathbf{x}, \hat{\mathbf{x}}, \boldsymbol{\phi})$.

Figure 4: Many-to-one accuracy of POS induction.

**Baselines.** We consider two baselines:

- hmm: a standard first-order hidden Markov model learned with EM;[13]
- fhmm: the directed alternative discussed in the main paper, as implemented by Berg-Kirkpatrick et al. [3], with the feature set from Smith and Eisner [38].

**Tuning Hyperparameters.** In our models, we use a squared $L_2$ regularizer for CRF parameters $\boldsymbol{\lambda}$, and a symmetric Dirichlet prior for categorical parameters $\boldsymbol{\theta}$ with the same regularization strength for all languages. The fhmm baseline also uses a squared $L_2$ regularizer for the log-linear parameters. The hyperparameters of our model, as well as baseline models, were tuned to maximize many-to-one accuracy for The English PennTreebank. The fhmm model uses $L_2$ strength = 0.3. The auto model uses $L_2$ strength = 2.5, $\alpha = 0.1$.

### A.2  Bitext Word Alignment Experimental Setup

Word alignment is an essential step in the training pipeline of most statistical machine translation systems [22]. Given a sentence in the source language and its translation in the target language, the task is to find which source token, if any, corresponds to each token in the target translation. We make the popular assumption that each token in the target sentence corresponds to zero or one tokens in the source sentence. Fig. 1 shows a Spanish sentence and its English translation with word alignments. As shown in Table 1 (right), an observation $\mathbf{x}$ consists of tokens in the target sentence, while side information $\phi$ are tokens in the source sentence. Conditioned on a source word, we use a categorical distribution to generate the corresponding target word according to the inferred alignments.

**Data.** We consider three language-pairs: Czech-English, Urdu-English, and Chinese-English. For Czech-English, we use 4.3M bitext tokens for training from the NewsCommentary corpus, WMT10 data set for development, and WMT11 for testing. For Urdu-English, we use the train (2.4M bitext tokens), development, and test sets provided for NIST open MT evaluations 2009. For Chinese-English, we use the BTEC train (0.7M bitext tokens), development, and test sets (travel domain).

**CRF Autoencoder Model Instantiation.** For word alignment, we define the reconstruction model as follows: $p_{\boldsymbol{\theta}}(\hat{\mathbf{x}} \mid \mathbf{y}, \phi) = \prod_{i=1}^{|\mathbf{x}|} \theta_{\hat{x}_i \mid \phi_{y_i}}$, where $\hat{x}_i$ is the Brown cluster[14] of the word at position $i$ in the target sentence. We use a squared $L_2$ regularizer for the log-linear parameters $\boldsymbol{\lambda}$ and a

Figure 5: Average inference runtime per sentence pair for word alignment in seconds ($y$-axis), as a function of the number of sentences used for training ($x$-axis).

symmetric Dirichlet prior for the categorical parameters $\boldsymbol{\theta}$ with the same regularization strength for all language pairs ($L_2$ strength $= 0.01$, Dirichlet $\alpha = 1.5$). The hyperparameters were optimized to minimize alignment error rate (AER) on a development dataset of French-English bitext. The reconstruction model parameters $\boldsymbol{\theta}$ are initialized with the parameters taken from IBM Model 1 after five EM iterations [6]. In each block-coordinate ascent iteration, we use L-BFGS to optimize $\boldsymbol{\lambda}$, followed by two EM iterations to optimize $\boldsymbol{\theta}$. Training converges when the relative improvement in objective value falls below $0.03$ in one block-coordinate ascent iteration, typically in less than $10$ iterations of block-coordinate ascent.

We follow the common practice of training two word alignment models for each dataset, one with English as the target language (forward) and another with English as the source language (reverse). We then use the grow-diag-final-and heuristic [23] to symmetrize alignments before extracting translation rules.

**Features.** We use the following features: deviation from diagonal word alignment $\left|\frac{y_i}{|\boldsymbol{\phi}|} - \frac{i}{|\mathbf{x}|}\right|$; log alignment jump $\log|y_i - y_{i-1}|$; agreement with forward, reverse and symmetrized baseline alignments of mgiza++ and fast_align; Dice measure of the word pair $x_i$ and $\phi_{y_i}$; difference in character length between $x_i$ and $\phi_{y_i}$; orthograhpic similarity between $x_i$ and $\phi_{y_i}$, punctuation token aligned to a non-punctuation token; punctuation token aligned to an identical token; 4-bit prefix of the Brown cluster of $x_i$ conjoined with 4-bit prefix of the Brown cluster of $\phi_{y_i}$; forward and reverse probability of the word pair $x_i, \phi_{y_i}$ with fast_align, as well as their product. The outputs of other unsupervised aligners are standard (and important!) features in supervised CRF aligners [4]; however, they are nonsensical in a joint model over alignments and sentence pairs.

**Baselines.** Due to the cost of estimating feature-rich generative models for unsupervised word alignment on the data sizes we are using (e.g., fhmm and dyer-11), we only report the per-sentence computational cost of inference on these baselines. For alignment quality baselines, we report on results from two state-of-the-art baselines that use multinomial parameterizations which support M-step analytic solutions, rather than feature-rich parameterizations: fast_align [15][15] and model 4 [6]. fast_align is a recently proposed reparameterization of IBM Model 2 [6]. model 4, as implemented in mgiza++ [17] is the most commonly used word alignment tool in state-of-the-art machine translation systems.

**Evaluation.** When gold standard word alignments are available (i.e., for Czech-English), we use AER [32] to evaluate the alignment predictions of each model. We also perform an extrinsic evaluation of translation quality for all data sets, using case-insensitive Bleu [33] of a hierarchical MT system built using the word alignment predictions of each model.

## Footnotes

[11]http://code.google.com/p/universal-pos-tags/

[12]We use brown-cluster -c 100 v1.3, available at https://github.com/percyliang/brown-cluster [25] with data from http://corpora.informatik.uni-leipzig.de/.

[13]Among 32 Gaussian initializations of model parameters, we use the HMM model which gives the highest likelihood after 30 EM iterations.

[14]We again use [25] with 80 word classes.

[15]https://github.com/clab/fastalign