[Reviews · NeurIPS 2014]

Submitted by Assigned_Reviewer_12

REPLY TO AUTHOR RESPONSE

Thanks for the response. You write: "It is possible we missed a related paper in the neural network literature. We would be grateful if you could provide citations of neural architectures which used hand-crafted features for unsupervised learning of structured outputs."

This may have misunderstood my point. I was really just making the same point as reviewer_26 (you responded to him or her).

But as it happens, I know of one branch of work loosely related to your question. Stoyanov et al. (AISTATS 2011, NAACL 2012) showed how to derive the structure of a sum-product feed-forward network by unrolling loopy sum-product belief propagation on a given graphical model. Each potential function in the graphical model can be parameterized in terms of hand-crafted features of its incident variables (e.g., it may be log-linear). These potentials convert into parametric weights of the network, reflecting the hand-crafted features. In addition, the structure of the network reflects the structure of the graphical model. The network is then trained by backprop to minimize prediction error at designated output variables. Is this "unsupervised"? Well, like the original graphical model, the resulting feed-forward network learns to predict the latent variables of the graphical model from the input variables in a way that is helpful for predicting the output variables. This could be regarded as "unsupervised" learning of a structure over the latent variables given the (input, output) variables. If the graphical model were given an auto-encoder structure where output = input, then their method would result in an unsupervised system very much like yours.

SUMMARY

Unsupervised learning of latent structure y given input x. This is a bottleneck method using an autoencoder p(x | y) p(y | x). Specifically, p(y | x) is a domain-specific CRF, while p(x | y) appears to be a simple independent model \prod_i p(x_i | y_i). The
y are compared with human annotations.

It is efficient to compute the objective (2) and its gradient. The objective is regularized log-loss of the reconstruction, marginalizing over y.

More generally, the autoencoder is p(\hat{x} | y,phi) p(y | \hat{x},x,phi) where \hat{x} is some ad hoc simplified version of x and phi is side information.

COMMENTS

The results are quite good. In particular, the method seems to produce fairly stable improvements across multiple languages (8 for POS tagging, 3 for alignment as evaluated by MT). However, the paper and supplementary material appear to have been written hastily. It is often hard to figure out the details of what was done experimentally.

The method is formally attractive and might turn out to be an important building block in learning certain kinds of latent structure. (Particularly if reconstructing the input x is only one task in a multi-task learning setting, so that y should also be useful to predict other, supervised properties of the input x. The authors don't discuss this.)

However, the authors don't do a good job of explaining why the method will do a good job of unsupervised learning. Their objective is to match human annotations, not just to learn representations that are useful for some other task. The usual difficulty in this setting is that the latent variables end up learning the "wrong" properties of the data -- not the properties that the humans annotated. The authors say something about this problem at L067, but they don't really explain why their method should do any better. Nor do they do any analysis of the experimental results to understand what was learned and why.

There are single-number comparisons with other methods, but no breakdowns or targeted experiments aimed at understanding what is going on. It's just left as a happy mystery. (There are no learning curves, either.)

It would also be helpful to explore whether the improved results are achieved because they reduce model error or reduce search error. For example, suppose method A (previous) and method B (this paper) find latent distributions p_A and p_B over the latent variables y. If each method is initialized separately at p_A and p_B, then does method A's objective prefer p_A and method B's objective prefer p_B, which implies that the objective is better? Or do they both prefer p_A or both prefer p_B, which suggests that the different stems from a search bias?

The authors suggest (L262) that their autoencoder architecture is more appropriate in some respects than neural autoencoders, although this assertion is not defended by discussion, empirical comparison, or error analysis. I think the authors are mainly motivated by the fact that the p(y | x) model can build on a long line of efficient, feature-engineered work on supervised structured prediction work in NLP. But there are many architectures that could use such features, including neural architectures.

DETAILED COMMENTS TO AUTHOR

The p(\hat{x} | y) model is described only obliquely. I have a good guess what it is, but please spell it out!

What exactly are you evaluating? At L172 you mention the distribution p(y | x,\hat{x},phi), under which, presumably, the probability of a given latent y is proportional to p(\hat{x} | y,phi) p(y | x,phi). Does this mean that you are evaluating the 1-best y from this distribution? One or many samples from this distrubtion? Something else?

It's not possible to really understand the features. For example, supplement L103 says that auto+full features include functions of the y_i, but supplement Table 2 doesn't show what those features are. It is also not clear whether "x_i, x_{i-1}" in the "full" column of that table is talking about conjoined features or is a list of features. And it's not clear whether the full model also includes all of the h&k features or not.

L147: Even before you introduced phi, you already allowed side information for the encoding phase, namely anything in x that was not part of \hat{x}. So perhaps you want to say that phi is side information that is (also) available in the reconstruction phase?

Possibly relevant is http://www.cs.cmu.edu/~nasmith/papers/gimpel+smith.naacl12b.pdf .

Suggestion on how to handle the supplementary material: Currently this is formatted as a separate paper, which is confusing. Please set it up as appendices to the main paper, following the bibliography, without duplicate material. This will allow you to have a single bibliography, a single set of figure and equation numbers, and crossrefs with the main paper. After running through pdflatex, use pdftk to divide the document into the main paper and the supplement.
Summary: This is an attractive method and the basic idea is quite appropriate for NIPS or ACL. The results appear strong. The downside is that there should be more analysis of why the method should work and why it does work (as well as a more careful description of the experiments, which I think we can trust the authors to fix).

Submitted by Assigned_Reviewer_20

The paper presents a CRF trained by auto-encoding through latent
variables y. (a) The P(y|x) is parameterized as in traditional CRFs.
(b) The regeneration P(\hat{x}|y) is a categorical distribution
independent for each \hat{x}_i and y_i pair. \hat{x}_i may be a
transformation of x_i, such as Brown cluster ids.

The paper is delightfully sensible and relatively simple (in a good
way). It falls into the category of papers that makes the reader say,
"Why hasn't this been done before; I wish I had thought of it."

I like it. The experimental results are positive, and I'm inclined
towards acceptance.

There is no reason this approach couldn't be used in a semi-supervised
setting. It would be great to see some results on these lines.

Along these lines, loosely related work that combines conditional and
generative training with log-linear models (but not necessarly in a
structured output setting) includes: (1) Tom Minka. Discriminative
models, not discriminative training. MSR-TR-2005-144, 2005. (2)
Andrew McCallum, Chris Pal, Greg Druck and Xuerui Wang.
Multi-Conditional Learning: Generative/Discriminative Training for
Clustering and Classification. AAAI, 2006.

Near the top of page 5 you describe alternative methods that would
require approximations on Z in the reconstruction phase. But I don't
believe you provide empircal comparisons with those methods. Are they
beaten by the other methods you do compare against?

On a related note, reading after Equation (5), and the "Feature HMM":
What do you loose by putting a "multi-feature view" of x only on the
P(y|x) side, but not on the P(\hat{x}|y) side? It would be nice to
have some discussion of this.

Section 3.2: You say the goal is being "coherent and interpretable".
Why is this the goal? You don't evaluate interpretability?

You picked just a subset of the languages in the CoNLL-X shared task.
This seems suspicious. How did you select them? Why not show all
languages?

Especially since scalability is an advertized advantage of your
approach, I would have liked to see a graph of test accuracy as the
amount of training data increases.

The objective function for this model is certainly not convex. It
would be nice to see some discussion of local minima, and the extent
to which you see empirial evidence of problems with respect to local
minima. What initialization do you use?

Minor writing issues:

Page 2: "offers two ways locations to impose" -> "offers two ways for
locations to impose" ?

"condition on on side information" -> "condition on side information"

Page 6: "and parsing Though" -> "and parsing. Though"

Summary: Well-written paper on a clean idea, with positive experimental
results. Relatively simple---in a good way---such that I expect it to
be used and have impact.

Submitted by Assigned_Reviewer_26

This paper presents an efficient way of doing unsupervised structured prediction by using a CRF as the first layer of an autoencoder. The paper is very clearly written, the idea is of the "I wish I had thought of this" kind, and the experimental results in POS tagging and MT alignment are solid and well-discussed. The relationship to previous work is presented fairly with one notable exception: the paper of Suzuki and Isosaki on semisup CRF training at ACL 2008. If you squint a bit, their proposal is actually rather similar to yours. In my own words (their paper is quite a bit harder to read than yours), what Suzuki and Isosaki do is to jointly train a CRF p(y|x) and a generative model p(x'|y) with a loss function that bounds a combination of the error on labeled data and the disagreement between the two models on unlabeled data. Their general approach is way more complicated than yours, and your presentation is leaner and more effective, but I feel that you need to give a careful account of the connection between the two approaches (and take care to look at more recent related work of theirs as well).

You claim that neural autoencoders cannot learn latent sequential structure without labeled data. That feels like an argument from ignorance. Maybe we/some of us don't know how to do that, but RNNs and LSTMs have at least the potential for doing it. I'd suggest a more measured comparison between these differen types of autoencoders.
Summary: A clearly written, convincing presentation of a simple but effective idea for unsupervised learning of structured predictors. I like this paper a lot, but the connection to some earlier work not cited needs to be sorted out.
Author Feedback
Author rebuttal: Thank you for the detailed reviews and helpful comments, especially suggestions for further analysis, pointing out gaps in experimental details, connections to related work, as well as formatting suggestions. We will do our best to incorporate these.

@reviewer_12:
- The ability to define hand-crafted feature representations to emphasize intended correlations is at the core of why we think the proposed method is better equipped to match human annotations than alternative unsupervised learning methods which don’t use features. We now realize this has been glossed over in the introduction. While we haven’t been able to derive a compelling theoretical argument, our experiments did confirm this intuition. Throughout the development of the proposed model, we ran controlled experiments to measure the impact of adding specific feature templates (e.g. suffixes and signature features for the POS induction task and deviation features for the word alignment task) which are known by domain experts to strongly correlate with human annotations. We also analyzed the model parameters associated with those features to make sure they are being effectively used. We will add further discussion about this in supplementary material. We will also consider adding targeted experiments and learning curves.
- It is possible we missed a related paper in the neural network literature. We would be grateful if you could provide citations of neural architectures which used hand-crafted features for unsupervised learning of structured outputs.
- All evaluations used the 1-best value of the latent structure according to the posterior p(y | x, \hat{x}, \phi). In our feature description, all features are conjoined with y_i, and observations x_i and x_{i-1} are independently conjoined with y_i. We will modify our description to clarify the confusion.

@reviewer_20:
- We are currently experimenting with semi-supervised learning, but it will be hard to squeeze it in this paper; we agree this is an important extension.
- In the part-of-speech induction task, the featurized HMM model which we use as baseline does outperform the undirected model (near top of page 5). In the word alignment task, the two alternative feature-rich models do not scale, according to personal experience as well as communication with the researchers who proposed the models for this task.
- In order to discover what we are missing by only using feature-rich representations in the encoding part but not the reconstruction part, we could either 1) use a logistic regression model to regenerate each observation in \hat{x}, or 2) use a (deficient) naive bayes reconstruction model which independently generates each feature of the observation conditional on the hidden structure. We have not done this comparison yet but we think this is an important question worth answering.
- While we don’t quantitatively measure model interpretability, it is important to be able to qualitatively assess model quality based on the parameters it learns and what they represent. We found this to be very helpful for debugging and analyzing the model.
- Before we start any experiments, this subset of the CoNLL datasets were picked for the convenience of using fewer and smaller datasets. In the final draft, we will report results on all datasets in CoNLL-06 and CoNLL-X.
- In preliminary experiments, we found that accuracy of our model for word alignment significantly improves as we use more data (this relationship has been observed for a long time for this problem). We will consider adding graphs to illustrate this relationship.
- Parameters of the reconstruction model are initialized with the emission parameters of a simple first-order HMM as discussed in the supplementary material. We use zero initialization of the CRF parameters.

@reviewer_26:
- Thanks for pointing out the related Suzuki and Isosaki ACL-08 paper; we will address this.
- Our claim in L266 that neural autoencoders cannot learn the latent structures of interest without training data was stronger than it needs to be. It is indeed conceivable that future work on neural autoencoders could do that. We will refocus this argument to point out current work with neural autoencoders has a different set of goals than learning structured representations.